# Effective Electrochemiluminescence Aptasensor for Detection of Atrazine Residue

**DOI:** 10.3390/s22093430

**Published:** 2022-04-30

**Authors:** Xue Huang, He Li, Mengjiao Hu, Mengyuan Bai, Yemin Guo, Xia Sun

**Affiliations:** 1School of Agricultural Engineering and Food Science, Shandong University of Technology, No. 266 Xincun Xilu, Zibo 255049, China; hxxh1985@163.com (X.H.); sdlihe88@163.com (H.L.); hu17852034056@163.com (M.H.); bai15695430950@163.com (M.B.); sunxia2151@163.com (X.S.); 2Shandong Provincial Engineering Research Center of Vegetable Safety and Quality Traceability, No. 266 Xincun Xilu, Zibo 255049, China; 3Zibo City Key Laboratory of Agricultural Product Safety Traceability, No. 266 Xincun Xilu, Zibo 255049, China

**Keywords:** atrazine, electrochemiluminescence, aptasensor, luminol, silver nanoparticles

## Abstract

According to the chemiluminescence characteristics of the luminol-hydrogen peroxide (H_2_O_2_) system, this work designed a novel and effective electrochemiluminescence (ECL) aptasensor to detect atrazine (ATZ) rapidly. Silver nanoparticles (AgNPs) could effectively catalyze the decomposition of H_2_O_2_ and enhance the ECL intensity of the luminol-H_2_O_2_ system. Once ATZ was modified on the aptasensor, the ECL intensity was significantly weakened because of the specific combination between ATZ and its aptamer. Therefore, the changes in ECL intensity could be used to detect the concentration of ATZ. Under optimal detecting conditions, the aptasensor had a wide linear range from 1 × 10^−3^ ng/mL to 1 × 10^3^ ng/mL and a low limit of detection (3.3 × 10^−4^ ng/mL). The designed aptasensor had the advantages of good stability, reproducibility, and specificity. The aptasensor could be used to detect the ATZ content of tap water, soil, and cabbage and had satisfactory results. This work effectively constructs a novel, effective, and rapid ECL aptasensor for detecting ATZ in actual samples.

## 1. Introduction

Triazine herbicides are widely used for preventing weeds in farmland, and one of the most representative is atrazine (ATZ, 2-chloro-4-ethylamino-6-isopropylamino-s-triazine). Due to its high efficiency and low price, ATZ is one of the most widely used herbicides to control broadleaf weeds and annual grasses that affect world agriculture [1,2]. ATZ has the characteristics of a stable molecular structure, durable effect, easy flow, and difficult degradation; thus, excessive use can cause residues in soil, water, and agricultural products [3]. ATZ residue remains in the soil for decades [4,5]. If human beings are exposed to an environment containing ATZ for a long time, the reproductive system, endocrine system, central nervous system, and immune system will be damaged even at very low concentration levels [6,7,8]. Moreover, the World Health Organization has classified it into carcinogen group 3, and long-term exposure increases the risk of breast and ovarian cancer in animals and humans [9,10,11]. ATZ residue in the soil can combine with some metals and soil humus to form some compounds [12], causing irreversible damage to the entire ecosystem. 

The European Union (EU), USA, China, and many other countries, have stipulated a limit on residue concentrations of ATZ in drinking water. In view of the significant toxicity of ATZ, it is very necessary to detect its residue in environmental samples precisely. Traditional methods for detecting ATZ residue mainly include gas chromatography (GC) [13], liquid chromatography (LC) [14,15], and gas chromatography-mass spectrometry (GC-MS) [16,17], liquid chromatography-mass spectrometry (LC-MS) [18], and other instrumental analysis methods. However, these methods require expensive instruments, complicated sample preparation, and time-consuming processes, so they are not beneficial for rapid on-site detection. Hence, it is necessary to develop rapid and on-site detection methods. Recently many papers have been published on the on-site detection of pesticides in agricultural foods, and many portable devices have been developed. This research has laid a foundation for the commercial application of pesticide residue on-site detection equipment in agricultural foods. For example, Kong et al. developed a fluorescent hydrogel test kit coordinated with a smartphone for on-site dimethoate residues analysis [19]. Zhao et al. constructed a smart plant-wearable biosensor for in situ pesticide residue analysis on crops [20]. Xie et al. used a plasmonic substrate with muli-hot spots to fabricate a highly sensitive SERS sensor to detect pesticide residues [21]. In order to realize the on-site detection of ATZ rapidly, it is first necessary to study the rapid detection methods. Currently, research on the detection of ATZ residues mainly include electrochemistry [22,23,24], surface plasmon resonance [25,26,27], immunoadsorption [28], surface-enhanced Raman spectroscopy [29], fluorescence [30,31], chlorophyll fluorescence imaging [32], chemiluminescence etc. [33]. Among them, electrochemistry is a dominant method, and most studies use antibodies and molecular imprinting as recognition elements. However, the essence of an antibody is protein, but it is volatile, and it requires strict operating temperatures, pH, etc., which is not conducive to operation and application. Studies have shown that ATZ antibodies cannot be closely distinguished from related molecules [34,35] therein the specificity of the immunosensor reduces. Molecular imprinting involves a multi-step synthesis and elution procedure and is thus more complicated. Williams et al. have successfully screened out an aptamer for ATZ [35]; however, there is little research on aptamer-based ATZ biosensors. When compared with antibodies and molecular imprinting, an aptamer is a single-stranded oligonucleotide that can specifically recognize targets, and it has the advantages of stable properties, effortless synthesis, and chemical modification.

Researchers mainly use electrochemical methods to study the rapid detection of ATZ, but electrochemical methods suffer from limited sensitivity [36]. ECL is an ultra-sensitive analysis technique that combines the advantages of electrochemistry and chemiluminescence [37,38,39,40]. Due to the advantages of a simple operation, wide detection range, high sensitivity, low equipment requirements, and low background interference, ECL is widely used in medical science, the pharmaceutical industry, environmental research, and other detection fields [41,42,43,44,45]. In recent years, ECL rapid detection technology has gradually become one of the research hotspots [46]. There are many reports that have been published on the detection of pesticides using ECL. Chen et al. used NH_2_-MIL-88(Fe) for an ECL aptasensor in a CdTe QDs-S_2_O_8_^2−^ system, which displayed superior sensitivity for the detection of malathion [47]. Liu et al. used luminol-gold nanoparticles-L-cysteine-Cu(II) composites (Lu-Au-Lcys-Cu(II)) to construct an ECL sensor for the detection of glyphosate based on a double inhibition strategy, which showed acceptable stability and selectivity [48]. Huang et al. constructed an ECL aptasensor by using copper (core)-gold (shell) bimetallic nanoparticles (Cu@AuNPs) for the detection of organophosphorus pesticides in a Ru(bpy)_3_^2+^-TPrA system, which had good stability and sensitivity [49]. However, there is only one report of using ECL to detect ATZ [50]. 

This work constructed a simple and sensitive ECL aptasensor to detect ATZ content rapidly, which was the first research on the detection of ATZ using an ECL biosensor with a chemiluminescent reaction between luminol and H_2_O_2_. The recognition element was an aptamer, and one end of the aptamer was modified with a sulfhydryl group. The work adopted AgNPs to enhance ECL sensitivity and intensity, and it also could connect with luminol and the aptamer through an Ag-S bond. 

The modified electrode accelerated the electron transfer and reduced the signal response time. Cyclic voltammetry (CV) and ECL analysis were conducted several times to estimate the current response and ECL intensity changes in the designed aptasensor. The designed ECL aptasensor had the advantages of sensitivity, economy, and high selectivity. Furthermore, the aptasensor was successfully used to detect the ATZ content in actual samples and had acceptable results. Compared with other ECL sensors for the detection of pesticides, the aptasensor also had good stability, sensitivity, reproducibility, and specificity; Although its limit of detection (LOD) was not the lowest, the construction of the aptasensor was simple, and the effect was good. Moreover, the aptasensor had the potential for application for use on-site and in real-time detection of ATZ residue in actual samples.

## 2. Experimental

### 2.1. Materials

Luminol was obtained from Aladdin Industrial Co. (Shanghai, China). AgNPs were offered by Nanjing XFNANO Materials Tech Co., Ltd. (Nanjing, China). ATZ, simazine, bromoxynil, paraquat, and malathion were obtained from A ChemTek Inc. (Morrisville, NC, USA). Propanil, 2,4-D were acquired from Alta Scientific Co., Ltd. (Tianjin, China). ATZ aptamer, and Bovine serum albumin (BSA) were received from Sangon Biotech Co., Ltd. (Shanghai, China), and the aptamer sequence was 5’-SH-(CH_2_)_6_-TGTAC CGTCT GAGCG ATTCG TACGA ACGGC TTTGT ACTGT TTGCA CTGGC GGATT TAGCC AGTCA GTGTT AAGGA GTGC-3’ [35].

During the experiment, concentrations of the main reagents were as follows: luminol was 0.05 mM, AgNPs was 0.01 mg/mL, BSA was 0.05%, PBS (Phosphate buffer solution) was 0.01 M, and H_2_O_2_ in PBS was 0.05 μM.

### 2.2. Apparatus 

Glass carbon electrodes (with a working diameter of 3 mm) were purchased from Tianjin Gaoss Union Photoelectric Tech. Co., Ltd. (Tianjin, China), which were used as the working electrodes. The CV analysis was undertaken through the CHI660D Electrochemical Workstation from Shanghai Chenhua Instrument Co., Ltd. (Shanghai, China) The ECL analysis was conducted on an MPI-A ECL workstation from Xi’an Remax Analysis Instruments Co., Ltd. (Xi’an, China) During the experiment process; both instruments used a three-electrode system with GCE (glass carbon electrode) as the working electrode, platinum as an auxiliary electrode, and Ag/AgCl as a reference electrode. Transmission Electron Microscopy (TEM)–Selected Area Electron Diffraction (SAED) and Scanning Transmission Electron Microscopy (STEM)–Energy Dispersive X-ray Spectroscopy (EDS) characterizations of AgNPs were conducted on the US FEI Tecnai G2 F20 transmission electron microscope. Ultrapure water was procured from a German Milli-Q water purification system.

### 2.3. Fabrication of the ECL Aptasensor

Figure 1 illustrates the fabrication procedure of the aptasensor, and the ECL aptasensor system was designed and prepared as follows.

Prior to the fabrication of the aptasensor, bare GCE was polished by 0.3 μm and 0.05 μm alumina powder slurries to achieve a mirror finish. After polishing, the electrode was ultrasonically cleaned for 30 s in ultrapure water, 1 min in absolute ethanol, and 30 s in ultrapure water, respectively, then dried naturally at room temperature for future use. Firstly, 5 μL of luminol solution was dropped on the surface of the electrode and allowed to dry naturally at room temperature. Secondly, 5 μL of the AgNPs solution was dropped onto the electrode surface. After drying, 5 μL of the ATZ aptamer was dropped onto the electrode surface. Then, 5 μL of 0.05% BSA solution was placed on the surface of the dried electrode to block the nonspecific sites; the BSA/Aptamer/AgNPs/Luminol aptasensor was obtained after drying. Finally, 5 μL of the ATZ solution was dropped onto the electrode surface to prove the performance of the fabricated aptasensor. The designed aptasensor was prepared to detect the concentrations of different ATZ solutions.

### 2.4. Experimental Conditions

In order to prove whether the aptasensor was constructed successfully or not, the construction process of the aptasensor was characterized by CV in 5.0 mM [Fe(CN)_6_]^3−/4−^ solution including 0.1 M KCl, at a scanning rate of 50 mV/s and a potential range of −0.1 to +0.6 V, which was used to monitor changes in electrochemical signals.

The ECL signals were obtained in PBS (pH 8.0, 0.01 M) using the CV mode with a continuous potential scanning range from 0 to 1.2 V, at a scanning rate of 0.1 V/ s. A high voltage of the photomultiplier was set as 700 V for the determination of luminescence intensity.

### 2.5. Samples 

In order to evaluate the potential application to detect the ATZ content of actual samples, tap water, soil, and cabbage were chosen as the detection samples. Tap water was obtained from our laboratory, soil from local farmland, and cabbage was purchased from a local supermarket. Tap water could be used to test directly without treating. The cabbage sample was processed as follows: Firstly, the cabbage was washed, dried, and ground successively, then 10 g of the cabbage homogenate was weighed. Secondly, 28 mL of methanol and 12 mL PBS was added (0.01 M, pH 9.0) and thoroughly mixed with ultrasound for 20 min. Thirdly, filter with filter paper to obtain filtrate; subsequently, the filtrate was centrifuged at 10,000 rpm for 5 min. Finally, the extracted supernatant was used as the sample solution. The treatment method for the soil sample was the same as the cabbage, except for the washing, drying, and grinding process.

All samples for the detection experiment were prepared through the spiking method. Three samples were spiked with ATZ standard solutions to make the concentrations of ATZ 100 pg/mL, 1 ng/mL, and 10 ng/mL. 

## 3. Results and Discussion

### 3.1. Characterizations of AgNPs

The TEM images of AgNPs are represented in Figure 1A–C. It shows that the AgNPs were spherical and had a good dispersion. The average particle size was about 15 nm. The crystalline nature of the AgNPs is shown by the SAED patterns in Figure 1D,E. It shows that the AgNPs had well distinguished diffraction spots. EDS Mapping analysis, as shown in Figure 1F–H indicated that there was only an Ag element and no other elements. The above results confirmed the successful synthesis of AgNPs.

### 3.2. Electrochemical and ECL Properties of the Fabricated Electrodes

Electrochemical analysis of the fabricated electrodes was performed by CV. After each fabrication step of the electrode, the working electrode response was analyzed.

According to Figure 2A, the bare glassy carbon electrode had a couple of obvious redox peaks because of the redox effect of [Fe(CN)_6_]^3−/4−^ (curve a). When luminol covered the electrode surface, the CV peak current increased significantly because luminol has good electronic properties (curve b). Then, AgNPs were dripped on the electrode surface, and the CV peak current increased dramatically, which was due to the good electron transfer and signal amplification effects of AgNPs (curve c). After the ATZ aptamer was dripped on the electrode surface, the current peak dropped significantly, which hindered the electron transfer between the reaction substrate solution and the electrode (curve d). When BSA covered the non-specific sites of the aptamer, the CV peak was further reduced (curve e). Finally, when ATZ covered the aptasensor, the peak current continued to decrease (curve f). The changes in the above CV peak current initially proved that the fabrication of the aptasensor was successful from the electrochemical point of view.

Following this, the ECL signal could further characterize the fabrication of the aptasensor. From (**B**), when luminol was dripped on the surface of the electrode, curve a showed that there was a clear ECL signal, indicating that luminol had a good ECL performance. Then, AgNPs were dripped on the electrode surface, and the ECL intensity further increased because AgNPs could catalyze the chemiluminescence of the luminol and H_2_O_2_ system and promote electron transfer (curve b). When the ATZ aptamer, BSA, and ATZ were dripped on the electrode surface in turn, BSA blocked the non-specific binding site of the aptamer, and the aptamer was specifically bound to the ATZ, which caused the ECL intensity to decrease sequentially (curve c–e). These phenomena existed because the modified substances on the electrode surface hindered the diffusion of the ECL co-reactant (H_2_O_2_) to the electrode surface, which formed a space barrier for electron transfer, and reduced the reaction between luminol and the co-reactant; thus, the ECL intensity gradually reduced. Frequently used AgNPs could effectively enhance the catalytic activity of ECL, and the concentrations of the experimental reagents were lower. In summary, the above results showed that the fabrication of an ECL aptasensor was successful.

### 3.3. Optimization of Experiment Parameters 

In order to obtain the best experiment conditions, some related experiment parameters were optimized, including aptamer concentration, the pH of PBS, incubation time, and scanning rate.

As shown in Figure 3A, when the concentration of aptamer was smaller than 100 nM, the intensity of ECL gradually decreased as the concentration of aptamer increased, and the intensity of ECL reached the minimum value at 100 nM of aptamer concentration. However, when the concentration of aptamer was greater than 100 nM, the ECL intensity gradually increased as the concentration of aptamer increased. This proved that the aptamer on the surface of the electrode reached a saturation state when the aptamer concentration was 100 nM. However, with a further increase in aptamer concentration, not enough aptamer could not be combined with AgNPs. Accumulation and entanglement phenomena might be formed on the electrode surface; thus, the ECL intensity increased. Hence, the best concentration of aptamer was determined as 100 nM to continue the subsequent experiments.

As seen in Figure 3B, the ECL of the Luminol-H_2_O_2_ system was influenced by different pH values of the reaction substrate solution. When the pH of PBS was smaller than 9.0, ECL intensity increased gradually as the pH of PBS increased. However, when the pH was greater than 9.0, the ECL intensity decreased gradually as the pH increased. It indicated that the ECL intensity reached the maximum value when the pH was 9.0, the ECL intensity was high and the sensitivity was enhanced in a relatively high alkaline solution, but too strong alkalinity hindered the luminescence. Thus, 9.0 was the best experimental pH value. 

Figure 3C shows the effect of incubation time on ECL intensity. After dripping ATZ, ATZ and the aptasensor were respectively incubated for 10, 20, 30, 40, 50, and 60 min. With the extension of incubation time, the ECL intensity increased gradually; however, the variation was not obvious after 30 min, which indicated that the ATZ had fully combined with the aptamer. As a result, the optimal incubation time was confirmed as 30 min. 

Figure 3D shows the relationship between the ECL intensity and the scanning rate. When the scanning rate did not exceed 100 mV/s, the ECL intensity increased with the scanning rate increasing and reached the maximum value at 100 mV/s. When the scanning rate exceeded 100 mV/s, the ECL intensity decreased as the scanning rate increased. The possible reason for this phenomenon was that intermediates generated slowly at a low scanning rate, and thus the ECL intensity was low. When a high scanning rate was used, the redox reaction time was shortened, and electron transfer was hindered; as a result, the ECL intensity decreased. Therefore, 100 mV/s was the best scanning rate.

### 3.4. ECL Analytical Performance of the Aptasensor

Under the best experimental conditions, the designed aptasensor could be used to detect different concentrations of ATZ solution. According to Figure 4, the ECL intensity decreased gradually as the ATZ concentration increased. 

When the concentration of ATZ solution ranged from 1 × 10^−3^ ng/mL to 1 × 10^3^ ng/mL, the ECL intensity and the logarithm of the ATZ concentration had a good linear relation. The linear equation was I_ECL_ = 3362.4 − 549LgC_ATZ_ (R^2^ = 0.998). The LOD (limit of detection) was 3.3 × 10^−4^ ng/mL (S/N = 3).

Compared with previous reported methods for detecting ATZ (Table 1), the designed aptasensor exhibited a wider linear range and lower detection limit. Except for the photoelectrochemical method, the LOD was lower than those of other methods in Table 1, but this work had a wider linear range than the photoelectrochemical method. This work successfully designed a simple and effective ECL method for detecting ATZ.

### 3.5. Stability, Reproducibility, and Specificity of the Aptasensor

In order to evaluate the stability of the designed aptasensor, under the same conditions, nine electrodes were prepared to detect 100 pg/mL of ATZ at different times. Among them, three modified electrodes were immediately dropped with ATZ, and then their ECL intensities were measured after 30 min of incubation time. The remaining six electrodes were stored in a dark experiment cabinet for detection, and three of them were measured every 5 days. Then, 5 days later, the detecting results showed that the ECL intensity of the three electrodes was reduced by 2.70% (RSD = 4.99%). After 10 days, the ECL intensity of the remaining three electrodes was reduced by 3.71% (RSD = 3.90%). The above results showed that the designed aptasensor for detecting ATZ exhibited good stability (Figure 5A).

Reproducibility was another important factor for the ATZ aptasensor. Then, four electrodes were prepared to detect 100 pg/mL of ATZ under the same experimental conditions. Their relative standard deviation (RSD) was 3.42%, which showed that the designed aptasensor for detecting ATZ had a good reproducibility (Figure 5B). 

This work chose six typical pesticides with similar structure or function as the interfering substances, including simazine, bromoxynil, 2,4-D, propanil, paraquat, and malathion, to estimate the specificity of the designed ECL aptasensor (Figure 5C). The results showed that the six single interfering pesticides without ATZ could not effectually hinder the ECL intensity under the same concentrations. Once the interfering pesticides contained ATZ, the ECL intensity would obviously decrease. The above results indicate that the designed aptasensor had a satisfactory specificity for detecting ATZ. 

### 3.6. Analysis of Actual Samples

In order to prove the potential application of the designed aptasensor, the aptasensor was used to detect the ATZ content of tap water, soil, and cabbage samples. The three samples were spiked with concentrations of 0 ng/mL, 0.1 ng/mL, 1 ng/mL, and 10 ng/mL ATZ, then recovery experiments were carried out on the samples. As shown in Table 2, the results showed that the recovery rate ranged from 89.13% to 123.03%. Moreover, all of the RSD was not more than 5.12%. The above results indicated that the designed aptasensor had high sensitivity and accuracy, which is promising for the detection of ATZ in an actual sample.

## 4. Conclusions

In summary, the study adopted the Luminol-H_2_O_2_ ECL system with AgNPs to successfully develop a highly sensitive and specific ECL aptasensor for the detection of ATZ. AgNPs could promote the decomposition of H_2_O_2_ to generate various intermediate reactive oxygen species and amplify signals, thereby enhancing the ECL intensity of luminol. Performing CV and ECL behaviors proved that the ATZ aptasensor was fabricated successfully. In addition, the designed aptasensor exhibited good stability, sensitivity, reproducibility, and specificity and had a low limit of detection. The designed aptasensor displayed good performance for the detection of ATZ within a linear range from 1 × 10^−3^ ng/mL to 1 × 10^3^ ng/mL, and the limit of detection was 3.3 × 10^−4^ ng/mL. The designed aptasensor can be used to detect the ATZ content of tap water, soil, and cabbage samples with satisfactory results. We believe that the aptasensor establishes a simple and rapid sensing method to enhance the ECL signal and shows potential for application in detecting ATZ residue in actual samples.

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
