# Peer review of "Effective Electrochemiluminescence Aptasensor for Detection of Atrazine Residue"

_sensors, 2022, doi:10.3390/s22093430_

Round 1

Reviewer 1 Report

Dear researchers,

                           This manuscript is excellent. but I think first that you have to put another one table in the end of the paper that writes the abbreviations. For instance ECL=...., GCE=...., e.t.c. Secondly, I believe that it needs little bit more to be written in introduction, I see that you just refer the previous work. This that it need it, is to find the previous job and describe what they do exactly, and then describe what did you do.  Thirdly after table 1, you must write a more descriptive view, why your LOD is better.

Finally, you should   put an error bar calibration curve.

This manuscript needs 1 or 2 more pages and more references in order to have a better scientific soundness. Acceptance after minor revision put it.

Kind regards

Reviewer 2 Report

The current manuscript entitled “Novel and Effective Electrochemiluminescence Aptasensor for Detection of Atrazine Residue” by “Huang et al” deliberated on the ECL aptasensor for the detection of Atrazine. The manuscript seems good and can be accepted after addressing the following comments.

  1. Remove the word novel from the title.
  2. In the abstract line 3 provides the full form of AgNPs.
  3. The authors should comment on the on-site detection of pesticide residues in agricultural foods. Recently many papers have been published on the on-site detection of pesticides. Many portable devices have been developed.
  4. Can the developed sensor be useful for on-site detection?
  5. In many places grammatical mistakes are there, please correct and revise thoroughly
  6. In the introduction authors mentioned, “Researchers mostly use electrochemical methods to study the rapid detection of ATZ, but its reproducibility and stability still need to be further improved.” How about the reproducibility and stability with the ECL technique?
  7. The authors mentioned the word “novel” to justify this in the introduction section.
  8. So many reports have been published on the detection of pesticides using ECL. Compare your works with a few other works in the introduction section.
  9. In the experimental section, the authors did not mention the SEM
  10. The graph is not clearly visible in scheme 1. Please revise.
  11. Provide the SAED pattern of AgNPs.
  12. Supporting characterization techniques are required for the confirmation of AgNPs.
  13. The authors should thoroughly revise the whole manuscript.

Round 2

Reviewer 2 Report

The revised manuscript can be accepted for publication.